# Neuroblastoma Risk Assessment and Treatment Stratification with Hybrid Capture-Based Panel Sequencing

**DOI:** 10.3390/jpm11080691

**Published:** 2021-07-22

**Authors:** Annabell Szymansky, Louisa-Marie Kruetzfeldt, Lukas C. Heukamp, Falk Hertwig, Jessica Theissen, Hedwig E. Deubzer, Eva-Maria Willing, Roopika Menon, Steffen Fuchs, Theresa Thole, Stefanie Schulte, Karin Schmelz, Annette Künkele, Peter Lang, Jörg Fuchs, Angelika Eggert, Cornelia Eckert, Matthias Fischer, Anton G. Henssen, Elias Rodriguez-Fos, Johannes H. Schulte

**Affiliations:** 1Department of Pediatric Oncology and Hematology, Charité—Universitätsmedizin Berlin, 13353 Berlin, Germany; annabell.szymansky@charite.de (A.S.); louisa-marie.kruetzfeldt@bihealth.de (L.-M.K.); f.hertwig@gmail.com (F.H.); hedwig.deubzer@charite.de (H.E.D.); steffen.fuchs@charite.de (S.F.); theresa.thole@charite.de (T.T.); stefanie.schulte@charite.de (S.S.); karin.schmelz@charite.de (K.S.); annette.kuenkele@charite.de (A.K.); angelika.eggert@charite.de (A.E.); cornelia.eckert@charite.de (C.E.); anton.henssen@charite.de (A.G.H.); elias.rodriguez-fos@charite.de (E.R.-F.); 2Division of Molecular Biology, Institut für Hämatopathologie Hamburg, 22547 Hamburg, Germany; heukamp@hp-hamburg.de (L.C.H.); willing@hp-hamburg.de (E.-M.W.); 3Department of Pediatric Hematology and Oncology, Universitätsklinikum Köln, 50937 Köln, Germany; jessica.theissen@uk-koeln.de (J.T.); matthias.fischer@uk-koeln.de (M.F.); 4The German Cancer Consortium (DKTK), Partner Site Berlin, 10117 Berlin, Germany; 5The German Cancer Research Center (DKFZ), 69120 Heidelberg, Germany; 6Experimental and Clinical Research Center (ECRC) of the Charité and Max-Delbrück-Center for Molecular Medicine (MDC) in the Helmholtz Association, 13125 Berlin, Germany; 7Berlin Institute of Health (BIH) at Charité—Universitätsmedizin Berlin, 10117 Berlin, Germany; 8NEO New Oncology GmbH, 51105 Köln, Germany; roopika.menon@siemens-healthineers.com; 9Department I—General Pediatrics, Hematology/Oncology, University Children’s Hospital, Eberhard Karls University Tuebingen, 72076 Tuebingen, Germany; peter.lang@med.uni-tuebingen.de; 10Department of Paediatric Surgery and Paediatric Urology, University Children’s Hospital, Eberhard Karls University Tuebingen, 72076 Tuebingen, Germany; joerg.fuchs@med.uni-tuebingen.de

**Keywords:** neuroblastoma, hybrid capture panel sequencing, *MYCN*, *ALK*, risk stratification, targeted therapies, NB-HCPS assay

## Abstract

For many years, the risk-based therapy stratification of children with neuroblastoma has relied on clinical and molecular covariates. In recent years, genome analysis has revealed further alterations defining risk, tumor biology, and therapeutic targets. The implementation of a robust and scalable method for analyzing traditional and new molecular markers in routine diagnostics is an urgent clinical need. Here, we investigated targeted panel sequencing as a diagnostic approach to analyze all relevant genomic neuroblastoma risk markers in one assay. Our “neuroblastoma hybrid capture sequencing panel” (NB-HCSP) assay employs a technology for the high-coverage sequencing (>1000×) of 55 selected genes and neuroblastoma-relevant genomic regions, which allows for the detection of single nucleotide changes, structural rearrangements, and copy number alterations. We validated our assay by analyzing 15 neuroblastoma cell lines and a cohort of 20 neuroblastomas, for which reference routine diagnostic data and genome sequencing data were available. We observed a high concordance for risk markers identified by the NB-HSCP assay, clinical routine diagnostics, and genome sequencing. Subsequently, we demonstrated clinical applicability of the NB-HCSP assay by analyzing routine clinical samples. We conclude that the NB-HCSP assay may be implemented into routine diagnostics as a single assay that covers all essential covariates for initial neuroblastoma classification, extended risk stratification, and targeted therapy selection.

## 1. Introduction

Neuroblastoma is the most common extracranial childhood tumor and is characterized by a broad clinical heterogeneity [1,2]. While low-risk neuroblastomas often differentiate into benign ganglioneuromas or spontaneously regress with no or minimal treatment, high-risk neuroblastomas often relapse despite the most aggressive multimodal treatment comprising chemotherapy, autologous stem cell transplantation, irradiation, and immunotherapy [3], and neuroblastoma relapse is almost always fatal. Despite the significant intensification of treatment over recent decades, high-risk neuroblastomas still account for 15% of all childhood cancer deaths. For many years, the risk-based therapy stratification of neuroblastoma patients has been the foundation of successful treatment, and it relies on clinical and molecular covariates [1,2,4,5,6,7]. Molecular covariates comprise amplification of the *MYCN* oncogene, and the structural chromosomal aberrations 1p36 loss, 17q gain, and 11q loss. In clinical routine diagnostics, these chromosomal aberrations are most often analyzed by fluorescence in situ hybridization (FISH). In recent years, next-generation sequencing (NGS) and systems biology analysis of neuroblastomas resulted in a better understanding of neuroblastoma tumor biology, the identification of additional molecular markers that improve risk assessment or define additional subgroups, and the identification of potential targets for molecular-targeted treatments of primary or relapsed neuroblastomas [8,9,10,11,12,13,14,15,16]. The discovery of rearrangements inducing the overexpression of TERT in the absence of *MYCN*-amplification [11,17] or the mechanisms of the alternative lengthening of telomeres (ALT), often due to mutation or deletion of *ATRX* [18], identified telomere maintenance as a feature defining high-risk neuroblastomas [11,13]. Activating mutations of anaplastic lymphoma kinase (ALK) or amplifications of the *ALK* gene, found in >10–15% of high-risk neuroblastomas, are correlated with adverse outcomes, but also define a potential target for a molecular-targeted therapy with ALK inhibitors [19,20,21,22,23,24,25,26]. Neuroblastomas with mutations in the RAS/MAPK pathway or the MDM2/TP53 pathway and active telomere maintenance were found to have only a minimal chance to be cured with current therapies, therefore being denominated “ultra-high-risk” (UHR) neuroblastomas [13,27]. Taken together, in order to translate current knowledge into the clinic, future clinical protocols for neuroblastoma treatment will require the analysis of many more molecular covariates than current protocols. While some programs implement whole-exome sequencing (WES) or whole-genome sequencing (WGS) and transcriptome sequencing into the analysis of mostly relapse tumors [28], these strategies are difficult to implement into the routine analysis of primary tumors, both due to the high costs and the need of high sample quality, which cannot always be ensured outside dedicated clinical trials. NGS panel sequencing analyzes specific genomic segments and a limited number of genes, and it is cost-effective and robust and, thus, applicable in clinical routine [29]. In contrast to early panel sequencing approaches that relied on PCR amplification, current hybrid-capture assays, in particular, minimize false negative results by reducing genetic noise, have a uniform coverage in the sequencing among samples, and can even be used to analyze formalin-fixed paraffin-embedded (FFPE) specimens [30,31]. The high sequencing depth achieved with panel sequencing can compensate for low tumor content or tumor subclonality in clinical samples. However, the applicability and clinical validity of panel sequencing assays heavily relies on the selection of genes and regions. For neuroblastomas, some pilot applications of panel sequencing have been reported to date [32,33,34]. Nevertheless, a clinical-grade neuroblastoma-specific sequencing panel is not available to date.

Here, we report the design, validation, and pilot clinical application of a custom-made, clinical-grade neuroblastoma hybrid capture panel sequencing (NB-HCPS) assay covering all genes and regions required for current and future risk stratification, tumor biology-based risk assessment, and the selection of molecular-targeted therapies in primary and relapsed neuroblastomas.

## 2. Materials and Methods

### 2.1. Patients and Samples

All patients were enrolled in the German neuroblastoma trial (NB2004) or the German Neuroblastoma Registry (NB Registry 2016) [6]. Written informed consent from patients or parents/guardians is available within the trial and registry documentation. Tumor samples were obtained from the German Neuroblastoma Biobank (Cologne). Whole-genome sequencing data and whole-exome sequencing data obtained within our previous studies were available for the 20 selected reference tumor samples (Appendix A) [11,13,14], and analyzed as previously described.

### 2.2. Cell Lines

The human neuroblastoma cell lines, IMR32, TR-14, BE(2)C, LAN-5, CHP-134, SK-N-DZ, KELLY, NGP, LAN-6, SH-EP, SH-SY5Y, SK-N-FI, CLB-GA, GI-ME-N, and CHLA-90, were cultured as described before [7,35]. In brief, cell lines were cultured in RPMI 1640 or DMEM (ThermoFisher Scientific, Waltham, MA, USA) supplemented with 10% fetal calf serum (GE Healthcare, Chicago, IL, USA), 1% penicillin (100 U/mL, ThermoFisher Scientific), and 1% streptomycin (100 μg/mL, ThermoFisher Scientific) at 37 °C and 5% CO_2_, before cells were harvested for subsequent DNA isolation. Master stocks for all cell lines were authenticated by short tandem repeat DNA typing by Idexx Bioresearch (Westbrook, ME, USA) or Multiplexion (Heidelberg, Germany). Whole-genome sequencing data of Kelly cells were made available from a previous study (SRA number: PRJNA622577) [36]. Reads were aligned to human reference assembly hg19 using Burrows–Wheeler Aligner MEM v.0.7.15 with default parameters; PCR and optical duplicates were removed with Picard v.2.16.0. Structural variation was performed using 5 different variant calling algorithms: novoBreak v.1.1.3 [37], SvABA v.1.1.1 [38], Delly2 v.0.7.7 [39], BRASS v.6.0.5 (https://github.com/cancerit/BRASS (accessed on 26 April 2021)), and SMUFIN v.0.9.4 [40] with default parameters. Due to the lack of a matched normal genome for Kelly cells, we used a healthy sample as a normal to perform the variant calling. Variants with breakpoints within a window of 500 bp were collapsed. Only intra- and interchromosomal variants supported by at least two different callers were taken into account. Breakpoints involving the *ALK:TERT* rearrangement were manually curated and reconstructed using the aligned reads from the BAM file and BLAT. Reads from RNAseq of the Kelly cell line [36] were aligned to human reference assembly hg19 using STAR aligner v.2.7.3a [41]; PCR and optical duplicates were marked with biobambam2 v.2.0.87 bamsormadup.

### 2.3. Hybrid Capture-Based Panel Sequencing

The NucleoSpin Tissue kit (Macherey-Nagel GmbH & Co. KG) was used to isolate DNA from neuroblastoma cell line pellets for hybrid capture-based panel sequencing. Tumor DNA was isolated from macrodissected cells from 10–15 10 μm sections of formalin-fixed, paraffin-embedded, or snap-frozen tumor material, and DNA was automatically extracted using a Maxwell Instrument and the Maxwell 16 FFPE Plus LEV DNA Purification Kit (Promega Corporation, Madison, WI, USA). DNA was sheared mechanically by ultrasonic acoustic energy (Covaris ultrasonicator (Covaris, Matthews, NC, USA). From each sample, 100–200 ng was subjected to our custom NB-HCPS assay (NEO New Oncology GmbH, Cologne, Germany). In brief, adapters were ligated to sheared sample DNA, and individual genomic regions of interest were enriched using complementary bait sequences (hybrid capture procedure). Selected baits ensured optimal coverage of all relevant genomic regions. After enrichment, targeted fragments were amplified (clonal amplification) and sequenced in parallel with an average mean sequencing depth of 1000–2500× (unfiltered reads). Sequencing data were processed and analyzed with NEO New Oncology’s proprietary bioinformatics pipeline [42,43]. Briefly, the quality of the raw sequencing reads was assessed, and the reads were aligned to the GRCh37 (hg19) human reference genome assembly. In order to detect translocations, discordantly aligned read pairs were subsequently extracted, and the structural variants and their breakpoints were localized based on reads spanning the breakpoint. Discordant reads were defined as paired reads, where mates had a mapping distance larger than the expected fragment size. Copy number alterations were detected by comparing the sequencing depth of the sample to the sequencing depth of technically matched control samples. The control samples were chosen to have no copy number alterations and a diploid genome, therefore only capturing the technical variation in the platform. In order to detect copy number alterations affecting genes or larger parts of a chromosome, the target region was subdivided into smaller bins, and ratios for each bin were computed by dividing the observed sequencing depth in the sample with the expected sequencing depth based on the control samples. Genes or larger regions were represented by several bins, where the majority of bins should show a ratio significantly deviating from one to support a copy number alteration for this gene/region.

## 3. Results

### 3.1. Design of a Neuroblastoma Panel Sequencing Assay

To establish a neuroblastoma hybrid capture sequencing (NB-HCPS) assay to be used at neuroblastoma primary diagnosis, as well as relapse, we aimed to include all regions relevant for international neuroblastoma risk stratification, as well as additional regions relevant for neuroblastoma tumor biology and targeted therapies. For that purpose, we screened the respective literature to select genes and regions to be included. We selected hybrid-capture probes to analyze copy number variations to detect amplifications of *MYCN*, chromosome 17q gain, and 1p36 and 11q loss, which are routinely analyzed at diagnosis to allow risk stratification according to current protocols using FISH or other appropriate methods [44]. To detect alterations associated with telomere maintenance, a crucial mechanism of neuroblastoma tumor biology, we selected probes to detect *ATRX* mutations or deletions [18], as well as *TERT* rearrangements [11,17]. Further probes were selected to detect *ALK* mutations or amplifications [19,20,45], focusing on the most common druggable mutations present in neuroblastoma, and genes involved in the p53/MDM2- and Ras/MAPK-pathway, as mutations of these genes have been previously shown to define a group of UHR neuroblastomas [13]. Finally, we included probes covering additional genes previously found mutated in neuroblastoma [9,10,11,12,13,18,46], and with potential relevance for neuroblastoma tumor biology or targeted therapies, including ATM [47] and ATR [48,49] (Appendix A). The NB-HCPS assay includes 4516 single probes that cover 2484 genomic regions, including 55 genes (Figure 1A, Appendix A), 736694 base pairs (bp) in total, as well as 65 additional probes across the genome serving as a reference for copy number analysis. Thus, with our NB-HCPS assay, we designed an assay covering all genomic regions and alterations relevant for international neuroblastoma risk stratifications, and additional regions relevant for neuroblastoma tumor biology and targeted therapies.

### 3.2. Analysis of Neuroblastoma Cell Lines

To test the performance of the NB-HCPS assay, we applied this assay to 15 neuroblastoma cell lines. The mapping of paired-end reads obtained by Illumina sequencing revealed that 44.8% (range 35.8–59.0%) of uniquely aligned reads could be mapped to the target regions (Appendix A), with a uniform coverage of target regions (Figure 1A,E, Appendix A). This resulted in a very high sequencing coverage (median coverage of 95% of target regions: 1367, range 212–3531; Figure 1E and Appendix A), even at a relatively moderate number of overall reads (median number of reads: 68,129,876, range 44,898,728–141,352,824; Appendix A), evidencing the efficiency of our approach. As expected, we were able to detect neuroblastoma-relevant copy number alterations (Figure 1B, Appendix A), including the exact determination of *MYCN* copy number. *MYCN* amplifications were detected in 8 out of 15 cell lines (Figure 1D, Appendix A). We detected the previously described *TERT* rearrangements in the cell lines CLB-GA and GI-ME-N, and we were able to identify a previously unrecognized TERT rearrangement in Kelly cells (Figure 1D) [13], which was not located in the typical genomic area of rearrangements, leading to overexpression of the *TERT* locus in neuroblastomas. In contrast, reconstruction using Kelly WGS data confirmed that the *TERT* rearrangement included a translocation between Chr.5 and Chr.2, with a breakpoint directly within the *TERT* gene, rather than in TERT regulatory regions. The breakpoint within the *TERT* gene and the translocation between Chr.5 and Chr.2 established a *ALK:TERT* fusion gene. The significance of this atypical rearrangement and previously undescribed fusion gene in Kelly remains elusive, as the presence of the *MYCN* amplification alone should be sufficient to explain the expression of the wildtype *TERT* allele and telomere maintenance in Kelly. In addition, single-nucleotide variants in the *ALK* gene were detected in 9 out of 15 cell lines (Figure 1D, Appendix A), including the F1174L mutation (in four cases) and the R1275Q mutation (in two cases), which are the most frequent ALK mutations in neuroblastoma [19,20]. While *ALK* was the gene most frequently harboring single-nucleotide variants (SNVs), a total of 33 SNVs were detected in 19 out of 55 analyzed genes. Analyzing cell lines, we not only confirmed that the NB-HCPS assay provided uniform coverage of the target regions, and allowed the detection of copy number alterations, rearrangements, and SNVs, but we also detected genomic alterations typical for high-risk neuroblastoma in a panel of 15 neuroblastoma cell lines.

### 3.3. Analysis of a Retrospective Neuroblastoma Cohort

To validate the NB-HCPS assay in primary neuroblastomas, we selected a retrospective cohort of 20 neuroblastomas (“reference cohort”), for which the results of routine reference diagnostic and WES or WGS were available (Appendix A). Note that the majority of selected cases were high-risk neuroblastomas, as high-risk neuroblastomas harbor more mutations. For all neuroblastomas, the presence of a *MYCN* amplification and chromosome 1p36 loss was determined by the reference laboratory of the German neuroblastoma study group using FISH analysis (Figure 2A). Using the NB-HCPS assay, samples were sequenced at a median coverage of 1150 (range 255–3391; Figure 2C, Appendix A). With detection of a *MYCN* amplification in 9 out of 20 neuroblastomas and the detection of 1p36 loss in 8 out of 20 neuroblastomas, results obtained with the NB-HCPS assay were in concordance with the results obtained by the reference laboratory, except for one case (Figure 2A,B, Appendix A). In this case, a 1p36 loss reported by the reference laboratory was not detected by the NB-HCPS assay. Note that FISH revealed a deletion of 1p36 in only 26% of the cells (2:1 in 14%, 3:1 in 10%, and 4:1 in 2%) in this case. Loss of 11q was detected in 2 out of 20 neuroblastomas, and 17q gain was detected in 5 out of 20 neuroblastomas (Figure 2A). *TERT* rearrangements were detected in 3 out of 20 neuroblastomas, and *ATRX* alterations were detected in 4 neuroblastomas (2 SNV and 2 losses). As observed in cell lines, the most frequently mutated single gene except for *MYCN* was *ALK*, with SNVs in 5 neuroblastomas and a gain in 1 neuroblastoma. In total, we detected 17 coding SNVs in 13 out of 55 analyzed genes (Figure 2A, Appendix A). We used WES/WGS data to validate SNVs detected with the NB-HCPS assay. The *CCND1* mutation in patient 16 was the only mutation detected by the NB-HCPS assay that could not be validated by WES/WGS, most likely because it was a subclonal mutation at a relatively low variant allele frequency of 10.5%, as detected by NB-HCSP. Note that separate samples obtained from the same tumor were analyzed with the NB-HCPS assay and WES/WGS in this case. Thus, the discrepancy in detecting the subclonal mutation could be explained by either spatial tumor heterogeneity, or by the relevant likelihood of WES/WGS not detecting a subclonal alteration at a 10.5% allele frequency with a local coverage of 45× in the respective case [50]. In addition, WES/WGS, which included the analysis of matching normal DNA, revealed that one SNV detected by the NB-HCPS assay (patient 1: *RICTOR*) was a germline SNV rather than a somatic mutation. Taken together, results obtained with the NB-HCPS assay in a retrospective reference cohort were in line with results obtained by the national reference laboratory and WES/WGS data, underlining the validity of the NB panel sequencing assay.

### 3.4. Prospective Routine Application of the NB-HCPS Assay

Subsequently, the NB-HCPS assay was used for the analysis of routine clinical samples (“routine cohort”), obtained as snap-frozen or formalin-fixed paraffin-embedded (FFPE) specimens. We analyzed a total of 81 neuroblastomas, and samples were collected at the primary biopsy (62 samples), at surgery after induction chemotherapy (9 samples), or at relapse (10 samples). Of the 81 specimens, 36 were fresh frozen and 45 were formalin-fixed. The median sequencing coverage was 1059 (range 103–3363, Figure 3C, Appendix A). The cohort of neuroblastomas was highly skewed toward high-risk, and in particular, high-risk neuroblastomas without *MYCN* amplification. In line, we detected 29 *MYCN*-amplified neuroblastomas (Appendix A), but also 42 *TERT* rearrangements and 8 *ATRX* alterations (Figure 3A). Surprisingly, several *MYCN*-amplified neuroblastomas also harbored *TERT* rearrangements. Subsequent analysis of the localization of the breakpoints of *TERT* rearrangements revealed that breakpoints could be located upstream of the *TERT* gene, or within the *TERT* gene (Figure 3D). In line with previous reports that reported partial *ATRX* losses more frequently than complete *ATRX* losses [18,51,52], 5 of 5 *ATRX* losses detected in our cohort were partial losses. Note that in 4 of 5 cases with partial *ATRX* deletion, at least one breakpoint was detected and reported as a rearrangement. In two additional samples, a breakpoint but no partial deletion was detected, most likely due to technical reasons (e.g., low tumor cell content leading to reduced sensitivity to detect deletions). As reported before, SNVs were more frequently in relapse, as compared to primary neuroblastomas (1.53 vs. 2.8, *p* = 0.3 (not significant)), but this trend did not reach significance in our cohort (Appendix A). *ALK* mutations were detected in 33 neuroblastomas, with R1275Q mutations in 14 and F1174L mutations in 10 cases. Based on the detection of mutations in the p53/MDM2 or Ras/MAPK pathways, 57 of the neuroblastomas would be considered ultra-high-risk in the presence of telomere maintenance mechanisms, according to the definition by Ackermann et al. [13]. In conclusion, we show here that the NB-HCPS assay could be used to analyze routine clinical samples providing information relevant for current and future risk stratification, as well as potential drug targets.

## 4. Discussion

While neuroblastoma risk assessment and therapy stratification have been complex ever since and traditionally relied on a combination of clinical and molecular risk factors, in recent years, NGS and systems biology have boosted the discovery of new genomic biomarkers. We report here the development and validation of the NB-HCPS assay, a DNA panel sequencing assay capable of providing all relevant genomic information required for neuroblastoma risk assessment, therapy stratification, and detection of molecular targets in a singular clinical applicable assay, which is the foundation to translate the recently described biomarkers into clinical practice.

While some previous panel sequencing studies were analyzing neuroblastomas, these assays were specialized for the detection of non-*MYCN*-amplified neuroblastomas [32], or genes relevant for risk stratification and the diagnosis of neuroblastomas such as *TERT* and *ATRX* were not included in the panel, or the setup of the assay was insufficient to detect chromosomal rearrangements and breakpoint regions [32,33,34].

In contrast, the NB-HCPS assay was specifically designed to provide all genomic information required for risk assessment and therapy stratification in current and future clinical trials. This includes the detection of SNVs, deletions, breakpoints/rearrangements, and amplifications, both of chromosomal DNA and of the recently discovered extrachromosomal circular DNA (ecDNA or eccDNA) [14,36,53]. Still, a major limitation is the assessment of telomere maintenance mechanisms, which cannot solely rely on genomic information. While the NB-HCPS provides robust information about *MYCN* status, *TERT* rearrangement, and *ATRX* mutations and deletions, ALT could be present also in the absence of *ATRX* alterations, and would then evade detection by the NB-HCPS assay. In addition, it is still a matter of debate if the analysis of telomerase expression should accompany genomic analysis. Thus, future molecular diagnostics of neuroblastoma will most likely include assays to analyze the ALT status and telomerase expression, in addition to genomic analysis. These analyses are of utmost relevance in neuroblastomas without *MYCN*-amplification, *TERT* rearrangement, and *ATRX* alterations, as these neuroblastomas are most likely low-risk, but the detection of telomerase overexpression or ALT would still render them high-risk.

Unexpectedly, we detected some cases in our cohort that harbored both *MYCN*-amplifications and *TERT* rearrangements. *MYCN*-amplification and *TERT* rearrangements were reported to be mutually exclusive in a previous study [11,13], although some overlap has even been reported in that original study [11]. The co-occurrence of *MYCN* amplification and *TERT* rearrangements detected in our study may be explained by intratumoral heterogeneities, or by the as yet poor definition of which type of *TERT* rearrangements lead to telomerase overexpression. For example, we detected a *TERT* rearrangement with a breakpoint within the *TERT* gene in Kelly, resulting in an *ALK:TERT* fusion gene of unclear biological relevance. It is unclear whether this breakpoint would result in the overexpression of functional telomerase, an expression of an *ALK-TERT* fusion gene of unknown functionality, or even a disruption of a functional *TERT* allele (compensated by overexpression of the wildtype allele driven by *MYCN*). In contrast, breakpoints of previously reported *TERT* rearrangements were located upstream or downstream of *TERT*, thereby affecting regulatory elements of the *TERT* genes and inducing the expression of wildtype telomerase [11,17]. It is tempting to speculate that two different types of *TERT* rearrangement exist, and it will be a challenge to analyze the biological relevance of *TERT* rearrangements with breakpoints that disrupt the *TERT* gene. For diagnostic purposes, it will be required to more exactly define the “classic” *TERT* rearrangement in a sense of Peifer et al. [11] and Valentijn et al. [17], which leads to the overexpression of wildtype telomerase and is a mechanism of telomere maintenance in neuroblastoma.

Note that the number of chr.1p36 and chr.11q losses detected in the prospective cohort was unexpectedly low. While the fact that the prospective cohort of neuroblastomas is highly skewed could contribute to the observation, it is not a sufficient explanation. In contrast to the reference cohort, the prospective cohort also included cases with lower sample quality or tumor cell content, which particularly complicates the detection of heterozygous chromosomal losses. In addition, variability in the region of Chr.1p36 or 11q loss and the presence of partial losses of the respective regions could further complicate detection. While we were not able to attribute the underdiagnosis of Chr.1p36 and 11q losses in the prospective cohort to a singular problem, we conclude that results of the NB-HCSP assay concerning the loss of chromosomal regions 1p36 and 11q must be interpreted with caution, particularly when applied to samples with a low tumor cell content or marginal quality. To improve detection of chromosomal alterations in future applications, we are in the process of implementing an improved version of the NB-HSCP assay that will contain a so-called SNP backbone, which evenly covers the genome with hybrid-capture probes at positions of common SNPs. The SNP-backbone will not only improve copy number calculation but also allow analysis of allele-specific copy number and, therefore, the detection of copy-number neutral losses of heterozygosity.

Panel NGS assays and, especially, hybrid-capture-based technologies have been successfully introduced to and approved for routine clinical use in other tumor entities [29,42,43,54]. Panel NGS applications, including the NB-HCPS assay, have several advantages that predestine them for clinical use. In particular, these assays are available at reasonable cost (and often are already reimbursed by health insurance), applicable to FFPE material and other low-quality material [30,31], and have an acceptable turnaround time with regard to wet lab processing, sequencing, and bioinformatics analysis [29].

While the ever-decreasing costs of WGS will allow broader application of WGS in the near future, hybrid capture-based sequencing will still outperform WGS with regard to coverage for a long time. A high coverage >1000× is important with regard to low tumor cell content and subclonality, both problems frequently observed in clinical samples.

It remains a matter of debate if, in addition to tumor tissue, panel sequencing assays such as the NB-HCPS assay should also include the analysis of matched normal control tissue, to discriminate between germline SNVs and mutations. In our study, we did not include the analysis of normal control tissue, which resulted in the detection of three mutations with unclear tumor/disease relevance in the reference cohort, which later turned out to be germline SNVs. This underlines the notion that the problem can be reduced if the panel sequencing assays only focus on relevant genes, and the analysis pipeline rigorously filters to exclude already-known SNVs and/or to include clinically relevant mutations. Nevertheless, in particular, the latter strategy could result in the nondetection of novel or rare but relevant mutations. Note that the ethical problem of detecting germline mutations is not present in panel sequencing assays focusing solely on tumor tissue, while strategies to deal with the detection of germline mutations need to be implemented when analyzing matched normal control tissue.

At the current stage, the NB-HCPS assay is ready to be tested in future clinical trials and will be even further improved by the application of a SNP backbone. As a first step, the NB-HCPS assay should be performed in parallel to the established biomarker assays, which the NB-HCPS assay will then replace in a second step. From the beginning, it will provide valuable information about new biomarkers, for which no traditional arrays have been established and evaluated.

## 5. Conclusions

We here present the NB-HCSP assay, which we designed to provide all molecular diagnostic information required for clinical risk stratification and application of targeted therapies for neuroblastomas in a single assay. We demonstrate feasibility of application of the NB-HSCP assay in a clinical setting. This is in line with hybrid capture based NGS panels being already in clinical use for adult cancers. The NB-HCSP assay may be implemented into routine diagnostics as a single assay, which covers all essential covariates for initial classification. In addition, the assay includes recently discovered risk factors for neuroblastoma (e.g. telomere maintenance, mutations within the RAS/p53 pathways), and in particular the ALK status, and therefore is able to identify a) patients that could be treated with ALK inhibitors and b) patients with ultra-high-risk neuroblastomas. The turn-around time of about a week supports the routine application of the NB-HCSP assay. While we aim to further improve the assay design by implementing a so-called SNP backbone, to further improve detection of chromosomal aberrations including LOH and heterozygous intrachromosomal losses, this improved version of the NB-HSCP, the improved NB-HCSP assay then warrants broad scale application and evaluation in a prospective clinical study.

## Figures and Tables

**Figure 1 jpm-11-00691-f001:**
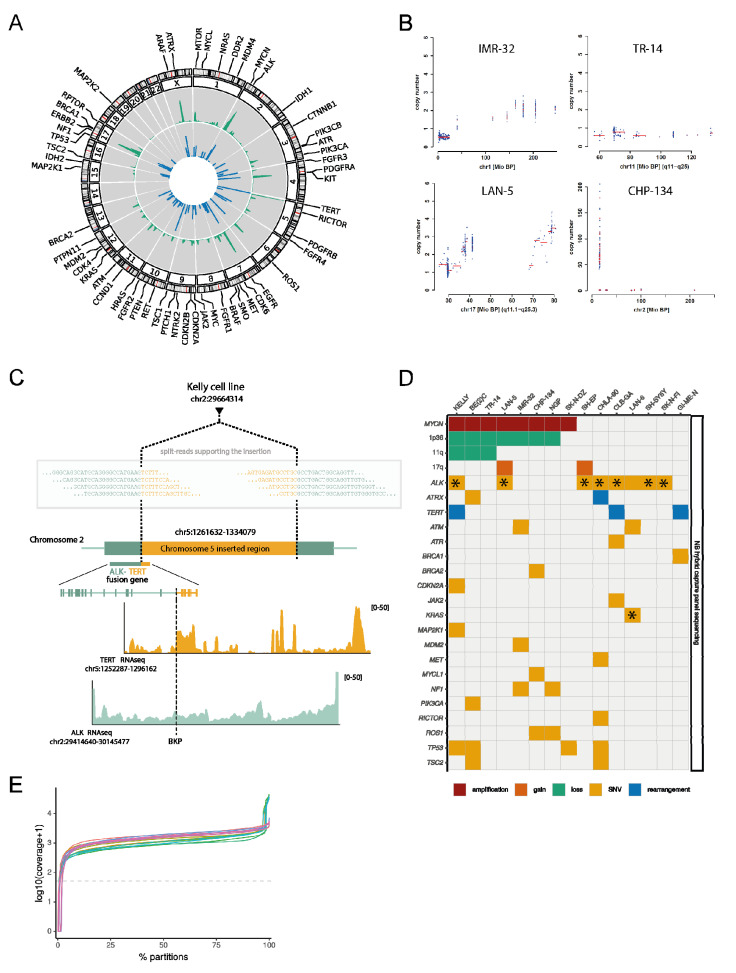
(**A**) Circos plot displaying the position of hybrid capture probes (blue ring) and read coverage of sequencing of the cell line Kelly (green ring). (**B**) Examples of copy number plots for chromosome 1 with 1p36 loss (top left, IMR-32 cell line), chromosome 11 with 11q loss (top right, TR-14 cell line), chromosome 17 with 17q gain (bottom left, LAN-5 cell line), and chromosome 2 with *MYCN* amplification (bottom right, CHP-134 cell line) based on data obtained with the NB-HCPS assay. (**C**) Location of the breakpoint supporting the insertion from chromosome 5 (*TERT*) into chromosome 2 (*ALK*), highlighting a fusion gene, which was detected in the cell line Kelly. (**D**) Oncoplot of alterations detected in cell lines. Mutations marked by stars (*) are highly likely to be involved in cancers (Tier 1), as reported in the Catalogue of Somatic Mutations in Cancer (COSMIC, Release v92). Note that amplifications, gains, losses, and rearrangements are displayed for MYCN, Chr. 1p36, 11q, 17q, ALK, and TERT, but for all other genes, only SNVs are displayed. (**E**) Sequencing coverage of all cell lines analyzed.

**Figure 2 jpm-11-00691-f002:**
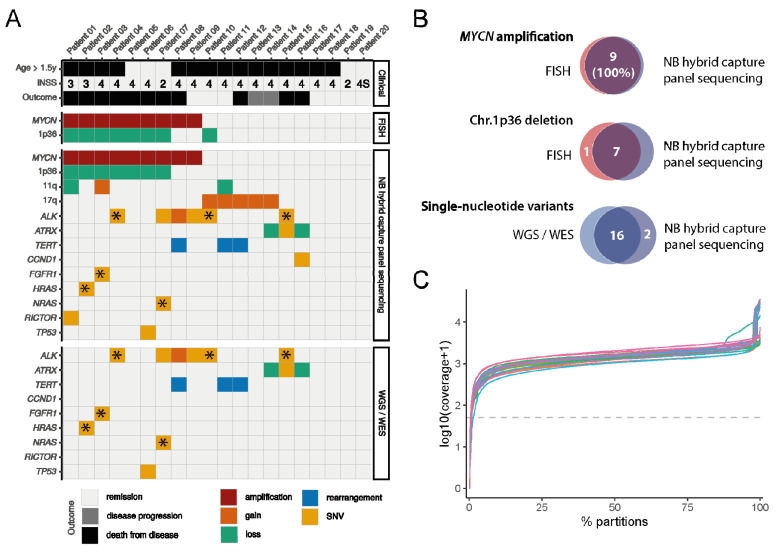
(**A**) Oncoplot displaying covariates and alterations detected with routine diagnostic FISH, the NB-HCPS assay, and WGS or WES in our neuroblastoma reference cohort. Mutations marked by stars (*) are highly likely to be involved in cancers (Tier 1), as reported in the Catalogue of Somatic Mutations in Cancer (COSMIC, Release v92), and were similarly found within the NB-HCPS assay and WGS or WES. (**B**) Venn diagrams displaying concordance of the NB-HCPS assay with routine diagnostic FISH and WGS/WES. (**C**) Sequencing coverage obtained with the NB-HCPS assay in samples of the reference cohort.

**Figure 3 jpm-11-00691-f003:**
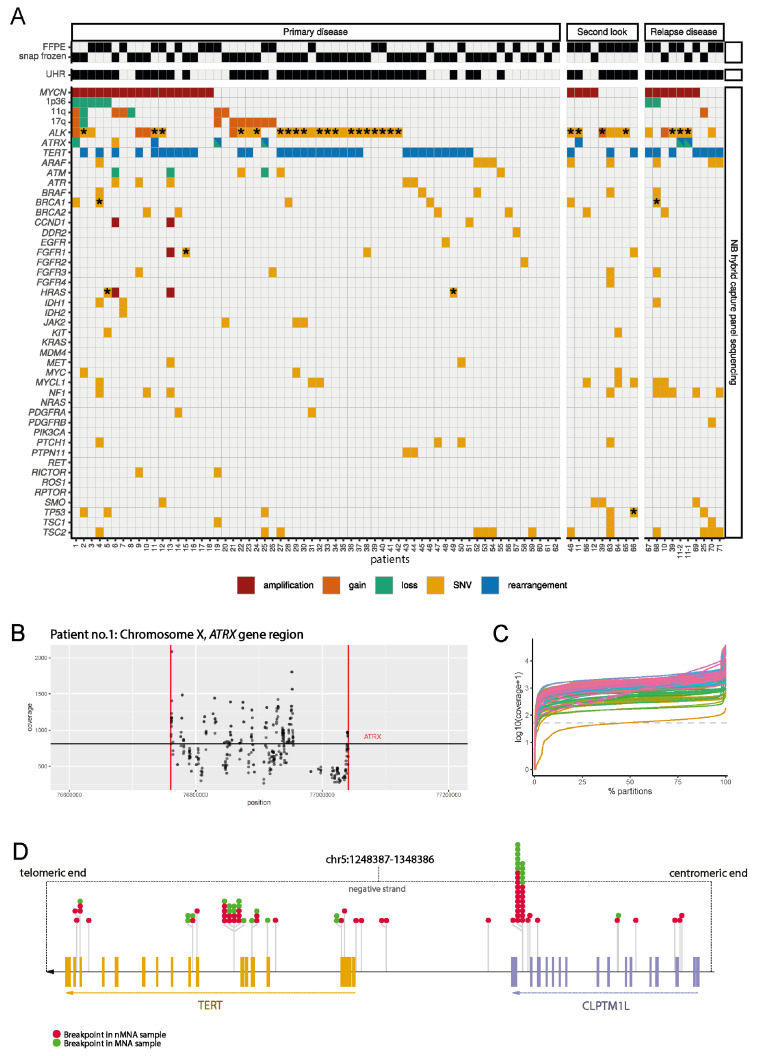
(**A**) Oncoplot of the cohort of routine samples displaying covariates and alterations detected with the NB-HCPS assay. Mutations marked by stars (*) are highly likely to be involved in cancers (Tier 1), as reported in the Catalogue of Somatic Mutations in Cancer (COSMIC, Release v92). (**B**) Example of partial ATRX loss detected in patient 1. (**C**) Sequencing coverage obtained with the NB-HCPS assay in samples of the routine cohort. (**D**) Distribution of breakpoints involved in the detected *TERT* rearrangements.

## Data Availability

WES Data were previously published in: Publication: PMID 30523111. European Genome-phenome Archive accession number (www.ebi.ac.uk/ega/, accessed on 14 October 2015): EGAS00001003244. WGS data were previously published in: Publication: PMID: 26466568. European Genome-phenome Archive accession number (www.ebi.ac.uk/ega/, accessed on 14 October 2015): EGAS00001001308.

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
