# Peer review of "Neuroblastoma Risk Assessment and Treatment Stratification with Hybrid Capture-Based Panel Sequencing"

_jpm, 2021, doi:10.3390/jpm11080691_

Round 1

Reviewer 1 Report

This is nice work trying to make it easier to obtain all of the genomic information necessary to care for patients with neuroblastoma.  You report the ability to identify the variety of genomic features detected by the panel, but does this validate with the known features of the cell lines or patient samples?  I agree that a prospective evaluation would be needed, but how does the information correlate so far?

Reviewer 2 Report

The authors of the manuscript titled "Neuroblastoma Risk Assessment and Treatment Stratification with Hybrid Capture-based Panel Sequencing" describe the application of the NB-HCSP assay in 20 NBs. This sequencing panel covers genomic regions and risk biomarkers already tested in routine NB diagnostics in a single assay. The manuscript is well structured and validation in clinical trials represents the future of molecular diagnostics of neuroblastoma. I have some observations for the authors:

  1. What are the criteria for choosing the 55 genes included in the panel. Do they look at the scientific literature?
  2. The authors are estimating the presence of somatic mutations without matching normal DNA. The authors assessed that this problem is minor because "the analysis filters for already known SNVs and / or clinically relevant mutations". This assessment is not in line with the panel design which includes the entire exons for each gene. They could impact in rare variants that would need to be filtered out. Most evidence reports that rare variants are very common in cancer genes. Somatic germline integration allows for the detection of rare variants that can influence individual susceptibility to tumor development.
  3. They assess that "the ethical problem of detecting germline mutations is not in the panel's sequencing assays that focus on tumor tissues." Incidental findings are mostly an ethical issue if whole exome/genome sequencing analysis is performed. The panel focuses on NB cancer genes, any incidental findings that could be a rare or common event should improve the quality of diagnosis, prognosis and therapy in personalized medicine. I think that using this panel with normal DNA germline maching will not generate ethical problems, because any genetic discovery could represent a fundamental step towards precision medicine (risk assessment and treatment stratification).

Reviewer 3 Report

The paper Szymansky et al. describes a novel assay for risk assessment in neuroblastoma.  The NB-HCSP panel covers the known neuroblastoma affected loci, oncogenes and could present a good tool neuroblastoma diagnostics. The retrospective study shows consistent results for the NB-HCSP assay compared to FISH and WGS/WES. The prospective cohort highlights that NB-HCSP can be used to detect MYCN amplifications and ALK mutations which are frequent mutations associated with high risk neuroblastoma. Also less common oncogenes are detected as amplified or mutated. However this cohort raises serious issues regarding the use for detection of loss of genomic regions.

Major point

 The prospective cohort of 62 neuroblastoma patients (Figure 3) identifies an unexpected low occurrence of loss at common loci for neuroblastoma:

*Chromosome 1p36 is usually lost in tumors with MYCN amplification, that is also seen in the cell lines and the retrospective cohort. In the prospective cohort 1p36 loss is only seen in 5 out of 18 primary cases.

*Chromosome 11q loss is frequently detected in neuroblastoma without amplified MYCN as described by Caren et al (PMID: 20145112) and others, is only found in 2 cases with MYCN amplification and not in any of the non-amplified cases.  Mandriota et al. (PMID: 26053094) identified ATM in the 11q loss regions. Figure 3 shows that ATM is lost in 3 cases, but not in the 2 cases annotated as 11q loss.

Figure 2B shows loss of chromosome 11q (q11-q25) in TR14, however ATM (11q22) is not identified as loss in panel 2D.

* ATRX mutations in neuroblastoma usually exists of internal deletions. In the prospective cohort of 62 tumors, no samples with ATRX loss are found.

The lack of identification of loss and conflicting ATM-loss/11q loss should be confirmed/discussed. Is NB-HCPS an adequate assay to detect loss? Is the material used for the retrospective cohort and the prospective cohort of different quality explaining this difference?

The NB-HCPS assay seems a good assay for amplification and mutation detection, but not for loss. If loss detection is a problem in the routine this will have implications, mainly to include ATRX in the risk assessment (which is one of the goals as described in lines 258-262).

The unexpected low level of loss in the routine samples urges an independent analysis to establish if this is an atypical patient group or if the assay cannot be used to detect loss in routine risk assessment.

Minor points

  1. CCND1 was identified in NB-HCPS and not by WES/WGS (lines 221-223). This could also be an amplification error. Was the mutation confirmed by amplification and sequencing?

  1. Figure quality should be improved.

Several panels are not or poorly readable:

 Axis (name and numbers) figure 2B

Figure 3C all text too small

  1. Figure 2A: Oncogenic mutations are indicted by * in the NB-HSCP panel and not in the WGS/WES panel (and are likely the same?)
